# The Optimal Operation and Dispatch of Commerce Air-Conditioning System by Considering Demand Response Strategies

Ching-Jui Tien, Chung-Yuen Yang ⬤, Ming-Tang Tsai * and Chin-Yang Chung

Department of Electrical Engineering, Cheng-Shiu University, Kaohsiung 833, Taiwan
* Correspondence: k0217@gcloud.csu.edu.tw; Tel.: +886-7-7310606

**Abstract:** The purpose of this paper is to discuss an optimal operation and schedule of commerce air-conditioning system by considering the demand response in order to obtain the maximal benefit; this paper first collects the operating data of the chiller units in commercial users, calculates the cooling load of each unit, and derives the relationship between the cooling loads and power consumption of each unit. The weather information, such as temperature and humidity of inside/outside, are collected in the EXECL database, and the cooling load of the mall's space is simulated by using the Least Square Support Vector Machine (LSSVM). Under the selected plan of power reduction, the requirement of space cooling loads, and the various operation constraints, the dispatch model of the commerce air-conditioning system with demand response strategies is formulated to minimize the total cost. A Modify Particle Swarm Optimization with Time-Varying Acceleration Coefficients (MPSO-TVAC) is proposed to solve the daily economic dispatch of the air-conditioning system. In the MPSO-TVAC procedure, the dynamic control parameters are embedded in the particle swarm of the PSO-TVAC in order to improve the behavior patterns of each particle swarm and increase its search efficiency in high dimensions. Different modifications in moving patterns of MPSO-TVAC are proposed to search the feasible space more effectively. By using MPSO-TVAC, it provides an optimal mechanism for variables regulated to increase the efficiency of the performing search and look for the probability of an optimal solution. Simulation results also provide an efficient method for commercial users to reduce their electricity bills and raise the ability of the market's competition.

**Keywords:** air-conditioning system; particle swarm optimization; demand response; least square support vector machine

## 1. Introduction

Energy, as the basic power to promote social development and economic activities, can improve the living quality and provide a convenient environment for human beings. In 2021, Taiwan's energy import dependency was 97%, and electricity consumption accounted for 49.4% of its final energy consumption. Stable energy and power supply are very important for Taiwan. Taiwan, as an island country, has to generate 100% of its own electricity, for its power system is not connected to the national power grids of other countries. Sufficient power supply has become the sincere expectation of people, industrialists, and businesspersons. In Taiwan, it is doomed that the future power supply will be changed, with thermal power accounting for 80% and renewable power accounting for 20%, which will inevitably lead to an increase in power generation costs. Accordingly, the electricity price will rise, which will impose considerable burdens on the electricity expenditures of enterprises. As a result, their power management technologies are important. Demand response plays an important role in power management strategies and can be adopted by power companies to postpone power development schedules, reduce temporal or regional line congestion, emergently relieve pressure from power rationing and provide mechanisms to maintain

safe power systems. Users with high electricity consumption reduce power demands by changing their electricity consumption habits and making proper use of the demand response mechanism. In practice, current business operators gradually pay attention to demand responses in order to achieve the most effective use of the annual electricity budget. Therefore, demand responses have become one of the key items in the power management strategies of business operators [1–3].

Because of industrial changes and the growth of air-conditioning electricity consumption in Taiwan, the difference between summer load and peak load and off-peak load continues to increase each year. In order to reflect the difference in power supply costs between peak load and off-peak load in summer and other seasons, Taiwan Power Company implements the system of season-of-use price and time-of-use price to reduce peak load and improve power supply; however, for general commercial or office buildings, the power consumed by central air conditioning systems may account for 40% of the total electricity consumption, indicating that they are usually the most power-hungry equipment [4,5]. Demand response (DR) is one of the strategies taken by power companies to reduce electricity consumption and provide incentives for electricity bill reduction during periods of tight power supply or high cost so that users reduce contracted load consumption [6,7]. In the demand response mechanism, excessively high agreed suppression contracted capacity may lead to a high reduction of basic electricity bills but will greatly improve the control of electric equipment. As a result, the agreed suppression contracted capacity even may not achieve, resulting in the failure of electricity bill reduction. Excessively low agreed suppression contracted capacity is easy to be achieved but is much less than the reduction of basic electricity bill. Without affecting their power demands or under normal operation, users should conform to the load management of the power company, reduce peak periods, improve participation willingness and implementation performance, get feedback and compensations, and improve power equipment utilization and power quality. Users with high electricity consumption reduce power demands by changing their electricity consumption habits and making proper use of the demand response mechanism. In practice, current business operators gradually pay attention to demand responses in order to achieve the most effective use of the annual electricity budget. Therefore, demand responses have become one of the key items in the power management strategies of business operators [8,9].

In recent years, there has been a lot of literature on the demand response scheduling and management strategies of the power market, mainly focusing on the effects of various operation strategies on the profits of power companies or users. Ref [10], with a virtual power plant as the bidder, calculates the demand response exchange in the day-ahead market by the stochastic programming method to obtain the maximum profit of the virtual power plant. Refs [11–13] take heating ventilation air conditioner (HVAC) as control load equipment to improve power quality and reduce electricity bills by implementing demand response optimization strategies. Ref. [14] carried out demand bidding by the time-based transferable load in and calculating users' best interests by automatic demand responses. In terms of electric energy management for domestic consumers, domestic air conditioning systems and energy storage systems are used with demand response policies of power companies to reduce users' electricity bills, increase extra incomes arising from electricity consumption reduction, and minimize electricity expenditures [15,16]. Some energy management control strategies are used to construct the demand response bidding model of integrated air conditioners, reduce load growth and evaluate users' profits so as to achieve the energy-saving dispatching of integrated air conditioners [17–19]; however, now, there is a trend of large public places domestically and internationally. For example, large public buildings such as commercial office buildings and shopping malls are springing like mushrooms. Large central air conditioning systems, usually the biggest electricity consumption equipment, are widely used in summer peak periods [20]. For these large power-consuming air conditioners used in buildings, some studies explore demand response strategies matched with energy management [21–23] and provide some effective management measures to reduce power supply bottlenecks. Ref [24] proposed

a supervised-learning-based approach to implement demand response control strategies for multi-regional building air-conditioning systems, seeking optimal regional energy management and integrating demand response control strategies at different electricity prices for maximum returns. Refs [25–27] integrated air-conditioning system scheduling, self-adaptive control strategy, and refrigerating capacity control in commercial buildings to provide reliable and stable demand response measures and effectively transfer peak load; however, in implementing demand response control of commercial buildings, especially shopping malls, people's comfort in the indoor shopping environment must be considered [28,29]. Ref. [30] proposed a comprehensive analysis of demand response pricing strategies in a smart grid environment. Most demand response periods are at peak load, and shopping malls are enclosed, crowded, and hot. Intermittent suspension of air-conditioning systems may cause inconvenience to people and reduce commercial transactions; however, in reality, due to the high power consumed by central air conditioning systems in large shopping malls, the demands for air conditioning should be regulated according to external climates and comfort indexes, and demand response measures should be coordinated, which leads to the increase of uncertain factors in electricity planning of shopping malls. Therefore, under the existing power system architecture, regardless of long-term or short-term scheduling and management, how to optimize economic and comfortable scheduling strategy by developing various electricity consumption control strategies will be a subject for commercial enterprises in sustainable operation.

The purpose of this paper is to discuss the optimal operation and schedule of the air-conditioning system by considering the demand response in order to obtain the maximal benefit. The weather information, such as temperature and humidity of inside/outside, are collected in the EXECL database, and the cooling load of the mall's space is simulated by using the Least Square Support Vector Machine (LSSVM) [31]. Under the selected plan of power reduction, the requirement of space cooling loads, and the various operation constraints, the dispatch model of the air-conditioning system with demand response strategies is formulated to minimize the total cost. A Modify Particle Swarm Optimization with Time-Varying Acceleration Coefficients (MPSO-TVAC) is proposed to solve this problem. In the MPSO-TVAC procedure, the dynamic control parameters are embedded in the particle swarm of the PSO-TVAC in order to improve the behavior patterns of each particle swarm and increase its search efficiency and accuracy in high dimensions. The proposed algorithm was tested in a shopping mall to prove its efficiency. Simulation results provide a novel method for commercial users to reduce electricity bills and raise the ability of the market's competition.

## 2. Problem Formulation

The system architecture studied in this paper is shown in Figure 1. Utility supplies power to the shopping mall to satisfy the load. Shopping malls then implement the demand response strategies in consideration of the comfort of people in the mall and the schedules of the air-conditioner system. With minimal cost as the goal, the optimal operation with demand response strategies is established to get the maximal benefits of a shopping mall.

### 2.1. Demand Response

Demand response is based on incentives provided by the electricity sellers to reduce electricity prices because power undersupply may occur during peak periods or power transmission congestion. If users are willing to reduce their electricity consumption, the system will change its power supply model and avoid the risk of power rationing. The model of planning electricity consumption reduction allows users to evaluate the business characteristics of enterprises, shopping malls, or factories and to sign electricity consumption reduction measures with electricity sellers to reduce electricity costs; this study took the demand response load management strategies currently developed to reduce electricity consumption by Taiwan Power Company, as shown in Table 1, including an 8-day monthly reduction model, 6-h daily reduction model, and 2-h daily reduction model [32].

The targeted subjects were users with high demand over 100 KW. The implementation period was from 1 June to 30 September each year.

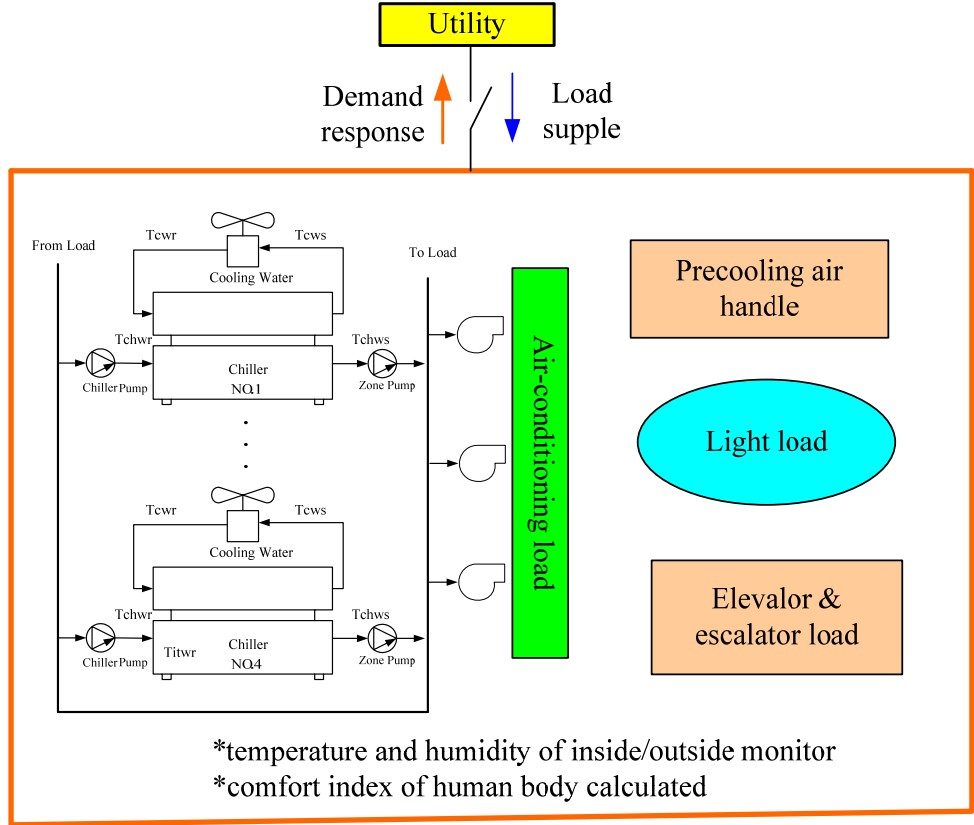

**Figure 1.** The system architecture studied in this paper.

**Table 1.** Planned electricity reduction measures.

| Planned Strategies | 8-Day Monthly Reduction Model | 6-h Daily Reduction Model | 2-h Daily Reduction Model |
|---|---|---|---|
| | 1 June to 30 September each year. | | |
| Suppression of electricity consumption period | From Monday to Friday every month, select the period of 8 days to reduce electricity consumption | From Monday to Friday every month | From Monday to Friday every month |
| | Appointment date 10 a.m. to 5 p.m. | Daily 10–12 a.m., 1–5 p.m. | Daily 1–3 p.m. |
| Suppression of electricity consumption time | 7 h | 6 h | 2 h |
| Scope of application | The targeted subjects were users with a high demand of over 100 KW | | |

Regardless of the type of planned strategies for electricity consumption reduction, the bill reduction was determined by the implementation rate ($x$). The implementation rate is defined as Equation (1).

$$x = \frac{Act_{contract}}{Agr_{contract}} \times 100\% \tag{1}$$

$x$: implementation rate.
$Act_{contract}$: actual reduced capacity.
$Agr_{contract}$: agreed reduced capacity.

There is a deduction rate ($y$) in the implementation rate ($x$) for each type of planned strategy, as shown in Tables 2–4 [32]. There is a corresponding deduction rate for electricity bills depending on the range of the implementation rate. The electricity bill reduced in the demand response strategies is calculated as in Equation (2).

$$DEB = BEP \times Agr_{contract} \times y \tag{2}$$

$DEB$: deduction of electricity bill.
$BEP$: basic electricity price.
$y$: deduction rate.

**Table 2.** The relationship between $x$ and $y$ in the 8-day monthly reduction model.

| Implementation Rate ($x$) | $x < 60\%$ | $60\% \leq x < 80\%$ | $80\% \leq x < 100\%$ | $x \geq 100\%$ |
|---|---|---|---|---|
| Deduction rate ($y$) | 0% | 10% | 20% | 30% |

**Table 3.** The relationship between $x$ and $y$ in 6-h daily reduction model.

| Implementation Rate ($x$) | $x < 60\%$ | $60\% \leq x < 80\%$ | $80\% \leq x < 100\%$ | $x \geq 100\%$ |
|---|---|---|---|---|
| Deduction rate ($y$) | 0% | 60% | 80% | 100% |

**Table 4.** The relationship between $x$ and $y$ in 2-h daily reduction model.

| Implementation Rate ($x$) | $x < 60\%$ | $60\% \leq x < 80\%$ | $80\% \leq x < 100\%$ | $x \geq 100\%$ |
|---|---|---|---|---|
| Deduction rate ($y$) | 0% | 30% | 40% | 50% |

*2.2. The Cooling Load Forecasting of Shopping Mall*

The temperature and humidity of the outdoor environment were obtained from the Central Weather Bureau Observation Data Inquire System (CWBODIS) [33], and the indoor temperature and humidity of commercial shopping malls were collected in the EXECL database. Based on the above information, the cooling load forecasting of the shopping mall was simulated by the LSSVM, as shown in Figure 2. Furthermore, body comfort was also considered in this research. The comfort index of the human body ($CI$) is one of the indexes to describe the comprehensive influences of temperature ($T$), humidity ($H$), and spatial environment wind velocity ($V$) on the human body, as shown in Equation (3) [29], indicating whether people feel comfortable at a certain temperature and relative humidity. Different combinations of temperature and relative humidity were used to represent the chilling capacity of an air-conditioned space.

$$CI = (1.818T + 18.18)(0.88 + 0.002H) + \frac{(T - 32)}{(45 - T)} - 3.2V + 18.2 \tag{3}$$

The cooling load of chillers in the conditioning system must meet the cooling load requirement of the shopping mall. The cooling load of chillers is generally calculated based on the return water temperature, supply water temperature, and the flow rate of chilled water. Therefore, the calculation of the cooling load capacity for the chillers is as in Equation (4):

$$Q_{chiller} = PLM \times \Delta T_{chw} \times \rho_w \times C_{pw} \tag{4}$$

$Q_{chiller}$: the cooling load of chillers.

*PLM*: the flow rate of chilled water (kg/s).
$\rho_w$: the density of chilled water (1 kg/L).
$C_{pw}$: the specific heat of chilled water (4.186 kJ/kg-°C).
$\Delta T_{chw}$ is the temperature difference of chilled water (°C), which is defined as Equation (5).

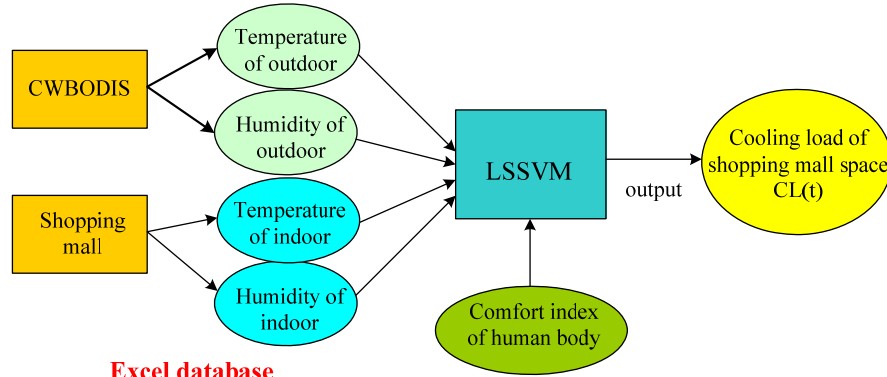

**Figure 2.** The cooling load forecasting of shopping mall by using the LSSVM.

$$\Delta T_{chw} = T_{chwrt} - T_{chwst} \tag{5}$$

$T_{chwrt}$: the return temperature of chilled water (°C).
$T_{chwst}$: the supply temperature of chilled water (°C).

The power consumption of chillers is a convex function of the cooling load capacity as shown in Equation (6):

$$P_{chiller,i} = a_i + b_i Q_{chiller,i} + c_i Q_{chiller,i}^2 + d_i Q_{chiller,i}^3 \tag{6}$$

where $a_i$, $b_i$, $c_i$ and $d_i$ are the regression coefficients of the function of cooling load capacity and power consumption.

### 2.3. Objective Function and Constraints

The main purpose of this paper is to derive the best single-day schedule planning for the air-conditioning system so that the total cooling load of the chillers can meet the required cooling load needs of the target space. By considering the demand response strategies, the total electricity cost can be formulated as in Equation (7).

$$\text{Min Cos}t = \sum_{t=1}^{H} \left( \left( \sum_{i=1}^{N} P_{chiller,i}(t)U_i(t) + SU_i(t) + SD_i(t) \right) \times TOU(t) \right) \\ - \sum_{t=1}^{T_1 \text{ or } T_2} DEB(t) \tag{7}$$

The constraints include both the system constraints and the unit's constraints:

(a)  load balance:

$$CL(t) = \sum_{t=1}^{H} \sum_{i=1}^{N} Q_{chiller,i}(t)U_i(t) \tag{8}$$

(b)  The limitation for the temperature difference of chilled water:

$$\Delta T_{chw,\min}^{i} \leq \Delta T_{chw}^{i}(t) \leq \Delta T_{chw,amx}^{i} \tag{9}$$

(c)  the limitation of the minimal operating time $T_{i,on}$/the minimal stopping time $T_{i,off}$

$$T_{i,on} \geq T_i^{on}, \ T_{i,off} \geq T_i^{off} \tag{10}$$

(d)   the limitation of cooling load for chillers

$$Q_{chiller,i,\min} \leq Q_{chiller,i} \leq Q_{chiller,i,\max} \tag{11}$$

$P_{chiller,i}(t)$: the power consumption of the i-th chiller at time t.
$U_i(t)$: the i-th unit on/off at time t, 1 is on and 0 is off.
$TOU(t)$: the TOU rates [34].
$SU_i(t)$: the power consume when the i-th unit is start at time t.
$SD_i(t)$: the power consume when the i-th unit is stop at time t.
$H/N$: the scheduling time/the total number of chillers.
$T_1/T_2$: the duration of demand response.
$DEB(t)$:Deduction of electricity bill at time t.
$Q_{chiller,i}(t)$: the cooling load of i-th chiller at time t.
$CL(t)$: the total cooling load of shopping mall at time t.
$T_i^{on}$: the maximal operating time.
$T_i^{off}$: the maximal stopping time.
$Q_{chiller,i,\min}/Q_{chiller,i,\max}$: the minimal cooling load/the maximal cooling load of the chiller.

## 3. Solution Algorithm

In a PSO system, birds (particles) flocking optimize a certain objective function. Each particle knows its current optimal position (*pbest*), which is analogous to the personal experiences of each particle. Each particle also knows the current global optimal position (*gbest*) among all of the particles in the population. Particle Swarm Optimization with Time-Varying Acceleration Coefficients (PSO-TVAC) is developed in [35]. Although PSO-TVAC can search in a wide range and has a high probability of obtaining the best solution, because there is no weight values, in the later search process, the regional optimal solution (*pbest*) and the global optimal solution (*gbest*) have no exchange ability. If the particles fall into the local optimal area, it still lacks the ability to escape from the local area; this paper proposes that MPSO-TVAC appropriately introduce the exchange mode of feasible and infeasible solutions of particles to increase its search ability and improve the ability of the population to search for particles in the whole area of the algorithm. The formulation of MPSO-TVAC is described as follows.

The velocity with PSO-TVAC can be represented in Equation (12). By using Equation (12), a certain velocity can be calculated due to the position of individuals gradually closer to *pbest* and *gbest*. The current position can be modified by Equation (13).

$$
\begin{aligned}
v_s^{t+1} = &\left[ c_1 = \left( c_{1f} - c_{1i} \right) \cdot \frac{iter}{iter_{\max}} + c_{1i} \right] \cdot rand \cdot \left( pbest_s^t - p_s^t \right) \\
&+ \left[ c_2 = \left( c_{2f} - c_{2i} \right) \cdot \frac{iter}{iter_{\max}} + c_{2i} \right] \cdot rand \cdot \left( gbest^t - p_s^t \right)
\end{aligned}
\tag{12}
$$

$$P_s^{t+1} = P_s^t + V_s^{t+1} \tag{13}$$

$c_{1f}, c_{2f}$: initial acceleration constant; in this paper, $c_{1f} = 0.8$ , $c_{2f} = 1.9$.
$c_{1i}, c_{2i}$: final acceleration constant; in this paper, $c_{1i} = 1.88$ , $c_{2i} = 0.7$.
$iter_{\max}$: the maximal iteration.
$iter$: the current iteration.
$rand$: uniform random value with a range of [0, 1].
$P_s^t$: the position of particle s at iteration t.
$V_s^t$: the velocity of particle s at iteration t.
$pbest_s^t$: the own best position of particle s at iteration t.
$gbest^t$: the best particle in the swarm at iteration t.

When PSO-TVAC is dealing with complex problems, it is very difficult to search for a feasible solution; it led to more trouble if the scheduling constraints are considered in the early stage of iteration. MPSO-TVAC introduces an operator, a "random feasible

solution", into the PSO-TVAC to increase the search ability. The "random feasible solution" process adds the proper random feasible own best position into the velocity vector when the solution is searched in each generation. MPSO-TVAC can be employed in the algorithm to make the search method more efficient at the end of the search, and the success rate of the search for a global optimum can be increased. The formulation of MPSO-TVAC is expressed as Equation (14).

$$v_s^{t+1} = \left[ c_1 = (c_{1f} - c_{1i}) \cdot \frac{iter}{iter_{\max}} + c_{1i} \right] \cdot rand \cdot \left( \text{pbest}_r^t - p_s^t \right) \\ + \left[ c_2 = (c_{2f} - c_{2i}) \cdot \frac{iter}{iter_{\max}} + c_{2i} \right] \cdot rand \cdot \left( gbest^t - p_s^t \right) \tag{14}$$

where $pbest_r^t$ is the own best position of random particle $r$ in all feasible particles at iteration $t$. Figure 3 shows the flowchart of the solution algorithm.

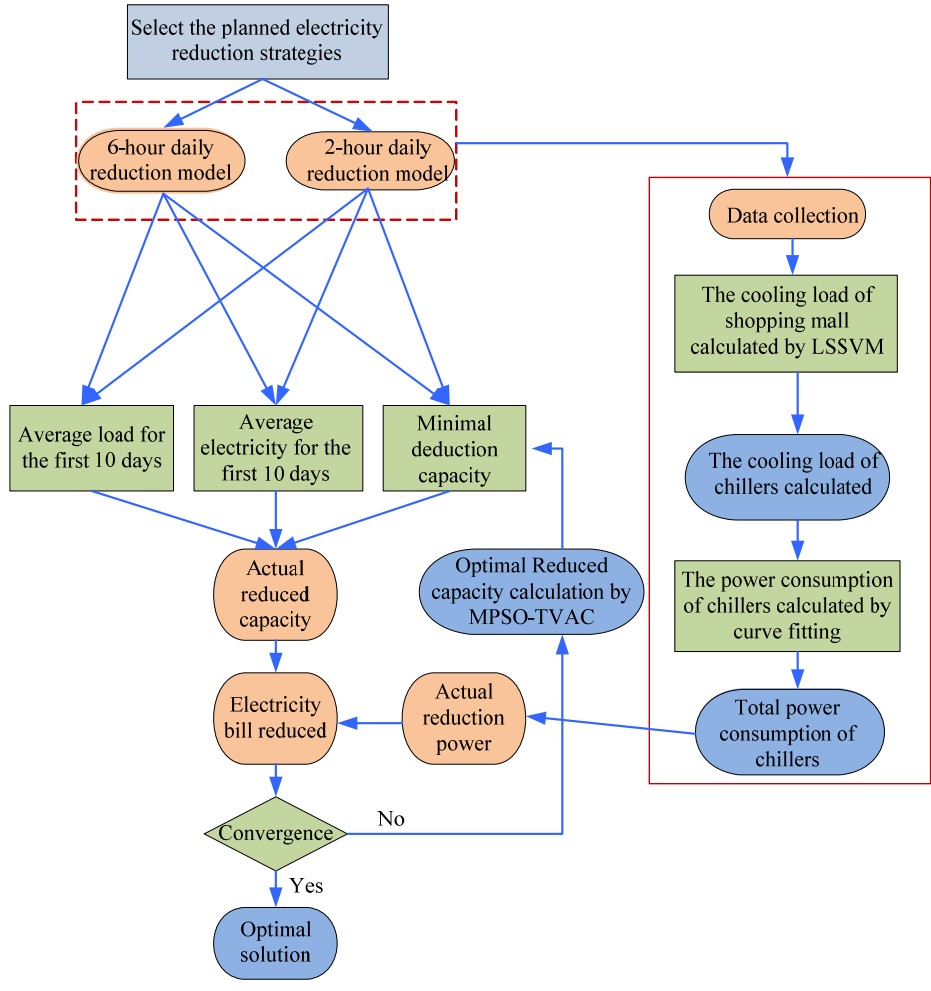

**Figure 3.** Flowchart of the solution algorithm.

## 4. Case Study

The proposed algorithm was tested in a commercial shopping mall. The shopping mall evaluated in this study covered an area of about 62,810 square meters and has roughly 170 stores. The mall had four large chiller units, three 860 HP chiller units, and one 560 HP chiller unit, totaling 3140 HP, to provide air conditioning for the mall. The electricity cost for the chillers was calculated based on the announced summer and non-summer prices from the TOU rate [32]. The temperature and relative humidity in the shopping mall were initially set as 26 °C and 45%, respectively. According to the electricity consumption model, the power demand rose greatly after 10:00 a.m.; moreover decline sharply after

10:00 p.m. In this study, Case 1 to Case 6 were designed based on the comfort index (CI). The temperature ranged from 26 °C to 28 °C, the relative humidity ranged from 45% to 55%, and the wind velocity was fixed at 1 m/s, indicating that the ambient temperature could make people feel warm and comfortable. Table 5 shows the corresponding parameters of the studied cases.

**Table 5.** The corresponding parameters of studied cases.

| Cases | Temperature (°C) | Relative Humidity (%) | CI |
|---|---|---|---|
| Case 1 | 26 | 45 | 72.337 |
| Case 2 | 26 | 55 | 72.350 |
| Case 3 | 27 | 45 | 73.977 |
| Case 4 | 27 | 55 | 73.990 |
| Case 5 | 28 | 45 | 75.621 |
| Case 6 | 28 | 55 | 75.635 |

### 4.1. The Power Consumption of Different Cases

Table 6 shows the power consumption of the different cases in a single period. The demand with different comfort index is calculated by the least squares support vector machine, as shown in Figure 4. Then, MPSO-TVAC is applied to optimize the scheduling of the air conditioning system. The total electricity consumption of the system is shown in Figure 5.

**Table 6.** The power consumption of the different cases in a single period.

| Case | Case 1 | Case 2 | Case 3 | Case 4 | Case 5 | Case 6 |
|---|---|---|---|---|---|---|
| maximal cooling load (RT) | 609.1354 | 593.4087 | 569.5376 | 563.4378 | 518.3609 | 488.8008 |
| maximal electricity consumption (kW) | 2086.093 | 2004.096 | 1965.254 | 1913.659 | 1757.882 | 1612.157 |
| total maximal electricity consumption (kW) | 6518.093 | 6483.879 | 6339.254 | 6292.916 | 5995.656 | 5986.157 |

### 4.2. 6-h Daily Reduction Model

In the 6-h daily reduction model of the planned strategies for electricity consumption reduction, the base contract capacity is 5383 kW and the basic bill is 223.6 NT/KW. In test cases with different CI, the optimal agreed reduced capacity in this study was calculated by MPSO-TVAC. Table 7 shows the simulation results of the 6-h daily reduction model. In test cases, the optimal agreed reduced capacity calculated by MPSO-TVAC was increased from 365 kW to 1221 kW, and the actual reduced capacity was increased from 365.024 kW to 976.949 kW, making the reduction of the basic electricity bill increase from 81,614 NT to 218,412.50 NT.

### 4.3. 2-h Daily Reduction Model

In the 2-h daily reduction model, the electricity consumption of the original system is used for evaluation. The Basic contract capacity from 10:00 a.m. to 12:00 p.m.; moreover from 3:00 p.m. to 5:00 p.m. on working days in August is 5042 kW. In the 2-h daily reduction model, the average power demand during the reduction period is from 1:00 p.m. to 3:00 p.m. As mentioned previously, the base power capacity is lower than the average power consumption during the reduction period. Hence, it is difficult to achieve the goal of the 2-h daily reduction model. Table 8 shows the simulation results of 2-h daily reduction model.

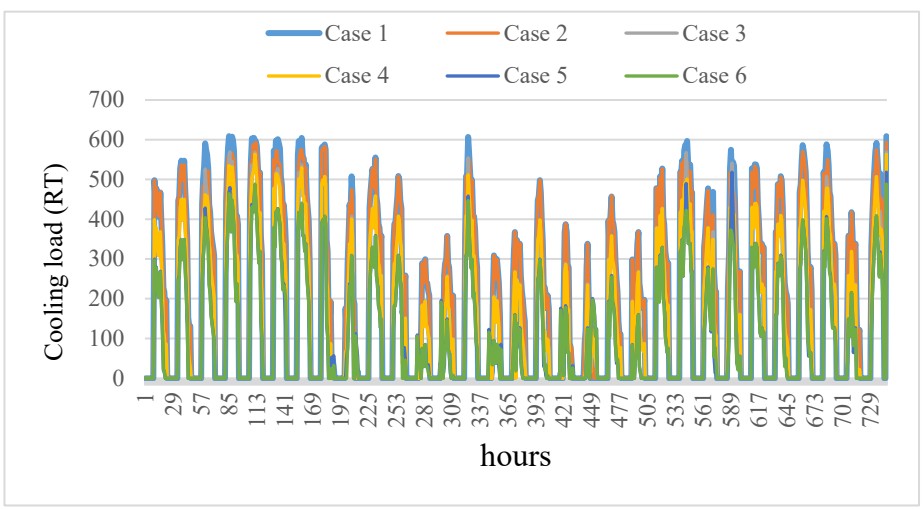

**Figure 4.** The cooling load of air-conditioner per hour in August 2020.

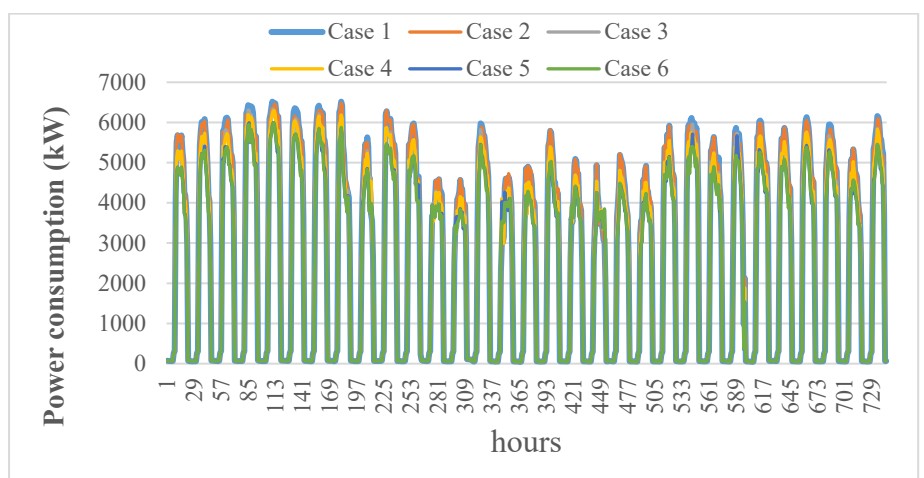

**Figure 5.** The power consumption of air-conditioner per hour in August 2020.

**Table 7.** The simulation results of 6-h daily reduction model.

| Cases | Basic Contract Capacity (kW) | Basic Bill (NT/kW) | The Average Power Consumption (kW) | Actual Reduced Capacity (kW) | Agreed Reduced Capacity (kW) | Implement Rate (%) | Deduction Rate (%) | Deduction Bill (NT) |
|-------|------|------|------|------|------|------|------|------|
| Case 1 | 5383 | 223.6 | 5018.142 | 365.024 | 365 | 100.01 | 100 | 81,614.00 |
| Case 2 | 5383 | 223.6 | 4981.512 | 401.655 | 502 | 80.01 | 80 | 89,797.76 |
| Case 3 | 5383 | 223.6 | 4718.968 | 664.199 | 664 | 100.00 | 100 | 148,470.40 |
| Case 4 | 5383 | 223.6 | 4702.757 | 680.409 | 1134 | 60.00 | 60 | 152,137.40 |
| Case 5 | 5383 | 223.6 | 4419.673 | 963.494 | 1204 | 80.02 | 80 | 215,371.50 |
| Case 6 | 5383 | 223.6 | 4406.217 | 976.949 | 1221 | 80.01 | 80 | 218,412.50 |

**Table 8.** The simulation results of 2-h daily reduction model.

| Cases | Basic Contract Capacity (kW) | Basic Bill (NT/kw) | The Average Power Consumption (kW) | Actual Reduced Capacity (kW) | Agreed Reduced Capacity (kW) | Implement Rate (%) | Deduction Rate (%) | Deduction Bill (NT) |
|-------|------|------|------|------|------|------|------|------|
| Case 1 | 5042 | 223.6 | 5414.284 | 0 | 0 | 0 | 0 | 0 |
| Case 2 | 5042 | 223.6 | 5350.107 | 0 | 0 | 0 | 0 | 0 |
| Case 3 | 5042 | 223.6 | 5099.841 | 0 | 0 | 0 | 0 | 0 |
| Case 4 | 5042 | 223.6 | 5066.662 | 0 | 0 | 0 | 0 | 0 |
| Case 5 | 5042 | 223.6 | 4797.349 | 244.775 | 407 | 60 | 30 | 27,301.56 |
| Case 6 | 5042 | 223.6 | 4782.205 | 259.920 | 433 | 60 | 30 | 29,045.64 |

In test cases, the optimal agreed reduced capacity calculated by MPSO-TVAC is increased from 0 kW to 433 kW, the actual reduced capacity is increased from 0 kW to 259.920 kW, and the reduction of the basic electricity bill is increased from 0 NT to 29,045.64 NT. If the internal ambient temperature is set above 28 °C, the actual reduced capacity will be greater than 0 in Case 5 to Case 6, so that the basic electricity bill can be reduced. Therefore, if the total electricity consumption of the system can be reduced overall, the 6-h daily reduction model in the planned strategies for electricity consumption reduction will be the most beneficial.

## 5. Conclusions

This paper takes a shopping mall enterprise as the point of view to determine the optimal operation and schedule of air-conditioning systems by considering the demand response; this paper proposed to use MPSO-TVAC to implement the comfort energy-saving control of the air-conditioning systems and the optimal agreement to reduce the contracted capacity under different conditions of air-conditioning load, comfort index, and demand response strategies. Operation planning, providing incentive feedback from the power company to obtain more benefit compensation, will be of great help to improve the use efficiency of its own power equipment and the ability of business operators to operate and manage. Through case tests, the planned strategies for electricity consumption reduction taken by the shopping mall under different comfort indexes were effectively evaluated. In the face of the power company's implementation of demand response, business users have a strategy to follow when they invest in energy-saving operation evaluation of air-conditioning systems to increase their own competitiveness and sustainable operation capabilities. Results are also shown that business users can obtain more benefit compensation from the power company to reduce electricity bills and obtain maximum operating benefits.

**Author Contributions:** C.-J.T. is the first author. He provided the project idea, related experiences, system model and revised English. C.-Y.Y. performed the experiments and conducted simulations. M.-T.T. assisted the project and prepared the manuscript as the corresponding author. C.-Y.C. contributed materials and tools. All authors have read and agreed to the published version of the manuscript.

**Funding:** This research was supported by the Ministry of Science and Technology, Taiwan (Grant Nos. MOST 111-2221-E-230-002).

**Institutional Review Board Statement:** Not applicable.

**Informed Consent Statement:** Not applicable.

**Data Availability Statement:** Not applicable.

**Conflicts of Interest:** The authors declare no conflict of interest.

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
