# Peer review of "The Optimal Operation and Dispatch of Commerce Air-Conditioning System by Considering Demand Response Strategies"

_inventions, doi:10.3390/inventions7030069_

Round 1

Reviewer 1 Report

The purpose of this paper is to discuss the optimal operation and schedule of the air conditioning system by considering the demand response to obtain the maximal benefit. The weather information, such as temperature and humidity of inside/outside, are collected in the EXECL database, and the cooling load of the mall’s space is simulated using the Least Square Support Vector Machine (LSSVM)[30]. Under the selected plan of power reduction, the requirement of space cooling loads, and the various operation constraints, the dispatch model of the air-conditioning system with demand response strategies is formulated to minimize the total cost.

A Modify Particle Swarm Optimization with Time-Varying Acceleration Coefficients (MPSO-TVAC) is proposed to solve this problem. In the MPSO-TVAC procedure, the dynamic control parameters are embedded in the particle swarm of the PSO-TVAC to improve the behavior patterns of each particle swarm and increase its search efficiency and accuracy in high dimensions. The proposed algorithm was tested in a shopping mall to prove its efficiency. Simulation results provide a novel method for commercial users to reduce electricity bills and raise the ability of the market’s competition.

Typo errors exist.

Equation styles and formatting are not correct.

What are the impacts of uncertainty on the results?

Literature is not complete. You can add more recent literature and compare your model with it. Example

https://doi.org/10.3390/math9182338

Images resolutions are not good.

Future works and limitations of the study to be enhanced/added.

Author Response

Dear Reviewer:

Thank you for providing us the review’s comments. We have taken care of these precious comments and revised manuscript with the changes clearly identified by a highlighter pen. The point-to-point responses to you are shown below.

  • Typo errors exist.

ANS: Typo errors had been corrected.

  • Equation styles and formatting are not correct.

ANS: Equation styles and formatting had been corrected.

3、Literature is not complete. You can add more recent literature and compare your model with it. Example https://doi.org/10.3390/math9182338

ANS: The recent literature had been added in Ref.[30].

4、Images resolutions are not good.

Ans.: Images resolutions had been improved.

I sincerely hope that we have clarified all your questions. Your assistance is very much appreciated. If you have further questions, please feel free to contact me.

Your assistance is highly appreciated.

Sincerely yours.

Dr. Ming-Tang Tsai

Department of E.E.,

Cheng-Shiu University

Email:k0217@gcloud.csu.edu.tw

Reviewer 2 Report

Justify why was particle swarm optimization (PSO) algorithm selected and modified over contemporary other Swarm-Based Optimizers which are mentioned in the reference below.  DOI is: https://arxiv.org/abs/2112.08421

Conclusion should be improved in terms of results obtained

What is the limitation of this model?

A white-box model can be provided for understanding the classification Refer following articles to lure potential readers and to develop a white-box model: DOI is: https://arxiv.org/abs/2112.08421 and https://doi.org/10.1115/1.4051696

Describe the layout of sensors used in CWBODIS.

What about the consideration of a change in temperature and humidity with respect to pollution, a toxic environment?

There are a few spelling mistakes, and grammar is improvable in a few places. There must be thorough proofreading of the paper.

Author Response

Dear Reviewer:

Thank you for providing us the review’s comments. We have taken care of these precious comments and revised manuscript with the changes clearly identified by a highlighter pen. The point-to-point responses to you are shown below.

1、Conclusion should be improved in terms of results obtained

Ans.: It had been added in “conclusion section”.

2、What is the limitation of this model?

Ans.:It had been described in “Problem Formulation Section”.

3、Describe the layout of sensors used in CWBODIS. What about the consideration of a change in temperature and humidity with respect to pollution, a toxic environment?

Ans.:It had described in “Section 2.2”.

3、There are a few spelling mistakes, and grammar is improvable in a few places. There must be thorough proofreading of the paper.

Ans.: The errors had been improved.

I sincerely hope that we have clarified all your questions. Your assistance is very much appreciated. If you have further questions, please feel free to contact me.

Your assistance is highly appreciated.

Sincerely yours.

Dr. Ming-Tang Tsai

Department of E.E.,

Cheng-Shiu University

Email:k0217@gcloud.csu.edu.tw

Round 2

Reviewer 1 Report

The paper can be accepted after the corrections made by the authors.